

**Wetlands inform how climate extremes influence surface water expansion and contraction**
Melanie K. Vanderhoof[1*], Charles R. Lane[2], Michael G. McManus[3], Laurie C. Alexander[4], Jay
R. Christensen[5]
[1]U.S. Geological Survey, Geosciences and Environmental Change Science Center, P.O. Box
25046, DFC, MS980, Denver, CO 80225
*email: mvanderhoof@usgs.gov, phone: 303-236-1411
[2]U.S. Environmental Protection Agency, Office of Research and Development, National
Exposure Research Laboratory, 26 W. Martin Luther King Dr., MS-642, Cincinnati, OH 45268
[3]U.S. Environmental Protection Agency, Office of Research and Development, National Center
for Environmental Assessment, 26 W. Martin Luther King Dr., MS-A110, Cincinnati, OH 45268
[4]U.S. Environmental Protection Agency, Office of Research and Development, National Center
for Environmental Assessment, 1200 Pennsylvania Ave. NW (8623-P), Washington, DC 20460
[5]U.S. Environmental Protection Agency, Office of Research and Development, National
Exposure Research Laboratory, Environmental Science Division, 944 E. Harmon Ave., Las
Vegas, NV 89119
**Abstract**
Effective monitoring and prediction of flood and drought events requires an improved
understanding of how and why surface-water expansion and contraction in response to climate
varies across space. This paper sought to (1) quantify how interannual patterns of surface-water
expansion and contraction vary spatially across the Prairie Pothole Region (PPR) and adjacent
Northern Prairie (NP) in the United States, and (2) explore how landscape characteristics
influence the relationship between climate inputs and surface-water dynamics. Due to differences
in glacial history, the PPR and NP show distinct patterns in regards to drainage development and
wetland density, together providing a diversity of conditions to examine surface-water dynamics.
We used Landsat imagery to characterize variability in surface-water extent across eleven
Landsat path/rows representing the PPR and NP (images spanned 1985-2015). The PPR not only
experienced a 2.6-fold greater surface-water extent under median conditions relative to the NP,
but also showed a 3.4-fold greater change in surface-water extent between drought and deluge
conditions. The relationship between surface-water extent and accumulated water availability



(precipitation minus potential evapotranspiration) was quantified per watershed and statistically
related to variables representing hydrology-related landscape characteristics (e.g., infiltration
capacity, surface storage capacity, stream density). To investigate the influence stream-
connectivity has on the rate at which surface water leaves a given location, we modeled stream-
connected and stream-disconnected surface water separately. Stream-connected surface water
showed a greater expansion with wetter climatic conditions in landscapes with greater total
wetland area. Disconnected surface water showed a greater expansion with wetter climatic
conditions in landscapes with higher wetland density, lower infiltration and less anthropogenic
drainage. From these findings, we can expect that shifts in precipitation and evaporative demand
will have uneven effects on surface-water quantity. Accurate predictions regarding the effect of
climate change on surface-water quantity will require consideration of hydrology-related
landscape characteristics including wetlands.
**Keywords**
Drought, evapotranspiration, Landsat, prairie pothole region, precipitation, surface water

**1. Introduction**

Surface-water dynamics have strong implications for ecosystem functioning and human

land use including biogeochemical balances (Hoffmann et al., 2009), species distribution
(Boschilia et al., 2008; Calhoun et al., 2017), hydrologic connectivity (Heiler et al., 1995;
Pringle, 2001)), and agricultural productivity (Mokrech et al., 2008; Gornall et al., 2010). Yet
natural variability in surface-water extent poses a basic challenge to gathering timely, accurate
information (Poff et al., 1997; Beeri and Phillips, 2007). While satellite imagery can be used to



map variability in surface-water extent over time, predicting future changes in surface-water
extent (e.g., in response to changes in climate, land use, or natural disasters) requires improving
our understanding of how the landscape influences surface-water extent over time and space. The
relative importance of hydrologic processes and flowpaths across a landscape (e.g., surface
storage, infiltration, evapotranspiration, runoff) can be expected to influence the timing, duration
and extent of surface water for a given location (Euliss and Mushet, 1996; LaBaugh et al., 1996,
van der Kamp et al., 1999)
Winter (2001) presented the concept of hydrologic landscapes as a means to classify
landscape units based on their hydrologic attributes (land-surface form, geology and climate).
These attributes, it is argued, could then be used to predict the partitioning of water into storage,
infiltration, evapotranspiration and runoff (Wagener et al., 2007). In many landscapes storage is
minimal and when rainfall intensity is greater than both the rate of soil infiltration and the soil
moisture deficit, runoff via overland and subsurface flows will dominate, contributing to
increased stream discharge (Eamus et al., 2006). These landscapes could be described as
exhibiting a low potential for surface-water expansion. Alternatively, in landscapes with low
topographic gradients and poorly developed drainage networks, runoff events rarely deplete
available surface storage, meaning that although stream discharge may elevate, much of the
surplus water remains as surface water (Shaw et al., 2012; Kuppel et al., 2015). These landscapes
show a high potential for surface-water expansion with evapotranspiration often the primary
mechanism for water loss (Winter and Rosenberry, 1998). Landscapes with a tendency to
accumulate surface-water are relatively common across the globe and include former glacial
landscapes including the Prairie Pothole Region (PPR) (Sass and Creed, 2008; Shaw et al.,
2012), and parts of China (Yao et al., 2007) and Russia (Stokes et al., 2007), permafrost regions





(Smith et al., 2007), as well as low gradient landscapes including the Argentine Pampas (Kuppel
et al., 2015); the Pantanal in Brazil (Hamilton, 2002), and the Orinoco Llanos in Columbia and
Venezuela (Hamilton, 2004). Although such landscapes have previously been shown to
experience surface-water expansion in response to increased precipitation (Huang et al., 2011;
Kuppel et al., 2015; Vanderhoof et al., 2016) or melting ice (Stokes et al., 2007; Yao et al.,
2007), we are unaware of studies that have explicitly compared surface-water expansion and
contraction between landscapes of differing surface-water expansion potential.

The PPR and adjacent Northern Prairie (NP), which span the upper mid-west of the

United States, occur within and beyond the last glacial maximum, respectively, and together
represent a range in the potential for surface-water expansion. The PPR is characterized by a
high density of depressional wetland and lake features (Zhang et al., 2009), a relic of glacial
retreat (Flint, 1971). Most wetlands are relatively small (< 0.5 ha) depressions, underlain by
glacial till with low permeability, and occur within a landscape matrix of natural grassland and
agriculture (Winter and Rosenberry, 1995; Zhang et al., 2009; Cohen et al., 2016). This is in
contrast to the adjacent NP such as the Northwestern Great Plains (Montana, western North and
South Dakota) and the Central Irregular Plains (southern Iowa and northern Missouri), which
lack the high density of small wetlands and show a well-developed drainage network due to its
occurrence outside of the last maximum glacial extent (USGS, 2013). The NP and PPR are also
characterized by substantial spatial and interannual variability in air temperature and
precipitation (Bryson and Hare 1974). Variations in temperature and moisture content of
competing air masses results in a strong north-south temperature and east-west precipitation
gradient. The precipitation-evaporation deficit is least in the east (i.e., Minnesota and Iowa), and
increases to the west (i.e., Montana) (Kantrud et al., 1989; Millet et al., 2009). This variability in



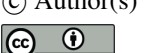

climate has a strong influence on water levels across the region. In the PPR in spring, wetland
depressions receive water from both precipitation and snowmelt. In the summer, water level is
controlled by direct precipitation, evaporation and wetland vegetation transpiration (Winter and
Rosenberry, 1995; LaBaugh et al., 1998; Carroll et al., 2005), with evapotranspiration typically
dominating water loss (Rosenberry et al., 2004).

Monitoring variation in water levels across the PPR has been of high interest as it is a key

factor in flood abatement, water quality, biodiversity, carbon management and aquifer recharge
(Gleason et al., 2008). Water level data at Devils Lake, North Dakota, for example, have been
collected as far back as 1867 and provide a regional indicator of hydrological conditions (Wiche,
1996; LaBaugh et al., 1996). Efforts have been expanded to map interannual changes in surface-
water extent across the PPR at a landscape scale using remotely sensed imagery (Kahara et al.,
2009; Niemuth et al., 2010; Vanderhoof et al., 2016). However, while substantial interannual
variation in water level has been documented across the PPR (Huang et al., 2011; Vanderhoof et
al., 2016), and primarily attributed to interannual variation in temperature and precipitation
(Johnson et al., 2005; Zhang et al., 2009), such surface-water patterns have to date been
minimally characterized for the remainder of the NP. In addition to interannual patterns of
temperature and precipitation, we would also expect that surface-water extent will depend on
landscape parameters such as infiltration capacity, storage capacity, and drainage characteristics
(Euliss and Mushet, 1996; LaBaugh et al., 1996; van der Kamp et al., 1999). Spatial models
incorporating some of these factors can provide additional insights into the risk of flood and
drought events across the PPR (Niemuth et al., 2010).

The PPR, in conjunction with adjacent NP, provides an ideal physiographic example in

which to analyze the influence of landscape characteristics on surface-water expansion and



contraction. Although the interaction between water level and climate has been studied
extensively at select locations within the PPR (e.g., Cottonwood Lake) (Winter and Rosenberry,
1998; Huang et al. 2011), minimal research has sought to understand spatial variability in the
relationship between climate and surface-water extent. Our research questions addressed in this
study are:
(1) How do interannual patterns of surface-water expansion and contraction vary

spatially across the Prairie Pothole Region and adjacent Northern Prairie of the

United States?

(2) How do landscape characteristics influence the relationship between climate inputs

and surface-water dynamics?

The successful exploration of this spatial patterning and landscape-scale statistical functions will
inform hydrologic/hydraulic and biogeochemical modeling and has implications for
biodiversity/habitat modeling and management (e.g., Allen et al., 2016; Golden et al., 2017)
**2. Methods**
As detailed below, we used Landsat imagery to map surface-water extent under dry,
average, and wet conditions across portions of the PPR and adjacent NP. We compared the
expansion and contraction of surface-water extent between the PPR and adjacent NP. As stream-
connected surface water can leave a location easily as stream flow, stream-connected and
disconnected surface water were analyzed separately. We then used a two-level modeling
approach to investigate the influence of landscape variables on surface-water dynamics. In the
first stage, surface-water extent per watershed was statistically related to accumulated water
availability, defined as precipitation minus potential evapotranspiration. This first stage produced
the dependent variable for the second model, the slope of the relationship between surface-water



extent and climate inputs per hydrological unit (a watershed) or the Surface Water Climate
Response (SWCR). The SWCR was then regressed against independent variables representing
landscape characteristics (e.g., infiltration capacity, surface storage capacity, stream density,
long-term climate normals). This approach allowed us to explore what landscape characteristics
drive spatial variability in the relationship between surface-water extent and climate.

**154    2.1 Study Area**

Our study area consisted of eleven Landsat path/rows (total area = 308,439 km$^2$) in the

U.S. portion of the PPR and adjacent NP (Figure 1). The PPR across North and South Dakota,
western Minnesota, northern Iowa and northern Nebraska, is dominated by the North and
Northwest Glaciated Plains. This ecoregion is characterized by landscape features formed during
its recent glacial history. Drift plains, large glacial lake basins and shallow river valleys support
row crop agriculture. Grasslands and livestock grazing dominate areas where glaciers left
deposits of uneven glacial till (Sayler et al., 2015). The PPR is dominated by cultivated crops
(59%), herbaceous (18%) and hay/pasture (10%) (Homer et al., 2015). Adjacent to the PPR, the
Northwestern Great Plains, across western North and South Dakota, is a semiarid unglaciated
plain which tends to have shallow soils with a clay texture not conducive to growing crops and
instead dominated by livestock grazing across grasslands (Sayler et al., 2015). To the southeast
of the North Glaciated Plains lies the Western Corn Belt and the Central Irregular Plains in Iowa
and Nebraska. Glacial till forms the parent material for most of the soil in Western Corn Belt and
the northern part of the Central Irregular Plains, within the study area. Level and gently rolling
hills and fertile soils support agriculture (Sayler et al., 2015). The NP is dominated by
herbaceous land cover (47%) with cultivated crops (28%) and hay/pasture (9%) is also common



171 (Homer et al., 2015). Using the precipitation averages (1981-2010) defined by the Parameter-

172 elevation Regressions on Independent Slopes Model (PRISM, Daly et al., 2008), the PPR study

173 area receives 6% more precipitation on average than the NP study area (626 mm yr$^{-1}$ relative to

174 592 mm yr$^{-1}$, respectively) and 1.5% less evaporative demand or potential evapotranspiration

175 (PET) (603 mm yr$^{-1}$ relative to 594 mm yr$^{-}$1, respectively). These differences were not found to

176 be statistically different using the Wilcoxon rank sum test.

177  Our regression analysis used eight-digit Hydrologic Unit Codes (HUC8s; USDA NRCS,

178 2015) as the unit of analysis (n=150) across all eleven Landsat path/rows (Figure 1). HUC8s

179 were used instead of smaller watersheds such as HUC10s or HUC12s to ensure that patterns in

180 surface-water expansion and contraction represented landscape patterns, not individual or small

181 groups of water features. HUC8s that occurred at the edge of a Landsat path/row with an area of

182 < 50 ha were excluded from further regression analysis to limit the inclusion of incompletely

183 characterized watersheds. The threshold of 50 ha was selected as it was a natural break in the

184 distribution of HUC8 sizes. Patterns of surface-water expansion and contraction were compared

185 between the PPR and NP. We note that one path/row (p37r26) in northern Montana was

186 technically within the most western section of the PPR, but was found to behave dissimilarly

187 from the PPR and similarly to the NP in terms of both its landscape characteristics (e.g., stream

188 density, wetland density) and surface-water expansion and contraction. Because of this, p37r26

189 was included in the adjacent NP for analyses where findings were organized by PPR and NP.


191 **2.2 Landsat Image Processing**

192 *2.2.1 Path-Row and Image Selection*





Surface-water extent was mapped for a series of images across 11 Landsat path/rows
(Figure 1). These path/rows were selected to represent the PPR and adjacent NP and were
intentionally selected to represent a range of ecoregions, climate conditions (west to east and
north to south) and densities of wetlands and streams. Snow-free images (acquired
approximately from April through October) containing less than 10% cloud cover from the
Landsat 4-5 TM, Landsat 7 ETM+ (prior to failure of the scan-line corrector in 2003) and
Landsat 8 OLI sensors were selected between 1985 and 2015. The number of images processed
within each path/row averaged 14 (range: 9 to 17 acceptable images) and were intentionally
selected to document interannual variability in surface-water extent, by selecting images from
wet, average and dry years (Table 1). The terms "wet", "average" and "dry" were defined in
reference to local norms, using the Palmer Hydrological Drought Index (PHDI) and the 12-
month Standardized Precipitation Index (SP12) (NOAA, NCDC, 2014). The range of conditions
captured by the time series within each path/row in relation to the historical climate conditions
(1895-2015) are shown in Table 1. The PHDI is based on a monthly water balance accounting
approach that considers precipitation, evapotranspiration, runoff and soil moisture. The indices
rely on weather station data and are interpolated at 5 km (NOAA NCDC, 2014). A complete list
of images included in the analysis is presented in the Appendix (Table A1).

*2.2.2 Image Processing*
Images were atmospherically corrected and converted to surface reflectance values using
the Landsat Ecosystem Disturbance Adaptive Processing System (Masek et al., 2006). A
minimum noise fraction transformation was applied to reduce within-image noise (Green et al.,
1988). The per-pixel water fraction was estimated using the Matched Filtering algorithm, a

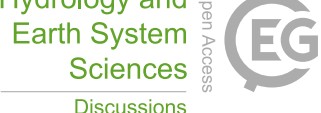

partial unmixing method in the ENVI software package (Exelis Visual Information Solutions,
Inc, Herndon, Va) (Turin, 1960; Vanderhoof et al., 2016). This algorithm is trained on a water
spectral signature, which was derived from open-water polygons manually selected within each
path/row, resulting in a water signature specific to each image. The water fraction output was
linearly stretched to maximize our ability to separate water from non-water. CFmask, a quality
control layer provided with Landsat images, was used to mask out clouds and cloud shadows
(Zhu and Woodcock, 2014), while the National Land Cover Database (NLCD) (2011) was used
to mask out impervious surfaces, defined as low, medium and high density development (Homer
et al., 2015), which can show spectral confusion with surface water. Each surface-water image
was visually inspected for quality using visual interpretation as well as ancillary datasets (e.g.,
National Agricultural Imagery Program (NAIP) imagery, National Wetland Inventory (NWI)
dataset (USFWS, 2010)). Select images were removed or edited primarily due to spectral
confusion between water and bare rock or shadowed vegetation.

*2.2.3 Surface-Water Extent Validation*
The surface-water extent maps were validated using 1-m resolution NAIP imagery.
Landsat images were selected for validation based on the temporal coincidence of the Landsat
and NAIP imagery collections (Table 2). Because terrestrial surface water is a relatively rare
cover type, it is difficult to generate enough inundated reference points through a simple random-
point generation. Therefore, random points were generated in reference to NWI polygons
overlapping with the NAIP and Landsat imagery. Points were then visually identified as
inundated or non-inundated using the NAIP imagery. To account for the scale difference
between a random point and a 900 m$^2$ Landsat pixel, the Landsat pixel boundaries for each



random point were identified. The point was classified as the majority class (inundated or non-
inundated) identified by NAIP within the Landsat pixel boundary surrounding each random
point. Reference points were generated per Landsat/NAIP pair (500 or 1000), with the number of
reference points varying depending on the amount of NAIP imagery available within the Landsat
path/row extent, and the number of random points that occurred within Landsat NA pixels.
Metrics presented included overall accuracy, omission error, commission error, dice coefficient,
and relative bias. Omission and commission errors were calculated for the category "water". The
dice coefficient is the conditional probability that if one classifier (product or reference data)
identifies a pixel as water, the other one will as well, and therefore integrates omission and
commission errors (Fleiss, 1981; Forbes, 1995). The relative bias provides the proportion that
water is under (negative) or overestimated (positive).

The Landsat per-pixel fraction water was binned into inundated ($\geq 0.3$) and non-

inundated ($< 0.3$) classes. This threshold was selected as it best balanced errors of omission and
commission. Overall accuracy for the Landsat surface-water maps across the 11 path/rows was
93.9% with errors of omission for surface water averaging 8.5% and errors of commission for
surface water averaging 8.2% (Table 3). The surface-water maps showed no relative bias and a
dice coefficient of 92%. Errors of omission and commission can be primarily attributed to mixed
Landsat pixels occurring over small wetlands (a few pixels in size) or at the edge of larger
wetlands or open water features. In some images parts of or entire agricultural fields were
classified as water. It is common in both the spring months, when crops need to be planted, and
fall months, when crops are being harvested, for fields to experience wet conditions (Fausey et
al., 1987; King et al., 2014). In addition, poorly drained soil is common across this region
(Skaggs et al., 1994) and wetland depressions often occur within agricultural fields.



Consequently, subsurface tile drainage has become increasingly popular across the region to
speed up the removal of excess soil water (Blann et al., 2009). It is often unclear to what extent
surface-water mapped within agricultural fields represents historical or current wetlands, poorly
drained fields, or misclassified pixels. Lastly, a close match in acquisition date between the
Landsat and NAIP images is essential for the NAIP imagery to accurately represent ground
conditions. Variability in the date match can be considered one potential source of error, as the
occurrence of a rain event or seasonal variability can change surface-water conditions over even
short time periods.

**2.3 Surface-Water Extent Analysis**

Surface-water abundance (ha km$^{-2}$) was calculated per HUC8 with HUC8 area being

adjusted for each image based on the abundance of not applicable (NA) pixels (e.g., cloud cover,
cloud shadow) in each image. We used the high-resolution National Hydrography Dataset (NHD,
1:24,000) to classify surface water as (1) continuous connected with the stream network, or (2)
disconnected from the stream network. The NHD line dataset was buffered by 14 m, the reported
digital horizontal accuracy of the dataset (USGS, 2000) and NHD area was added to account for
the width of large rivers. Surface-water polygons that intersected the stream network in a given
image were classified as continuously connected water (CCW). Surface-water polygons that did
not intersect the stream network in a given image were classified as discontinuous water (DCW)
or discontinuous from the stream network. We acknowledge that the NHD is known to be
incomplete (e.g., lacking short and ephemeral stream lines) and that some stream lines within the
NHD are disconnected from downstream waters (Heine et al., 2004). However, the NHD is the
most complete nationally-available stream dataset.





Processed images within each path/row were ranked from least-to-most amount of
surface water per area. Median condition was defined as the image or two images representing
the median amount of surface-water extent, estimated from all images within a path/row.
Drought and deluge conditions were defined as the average of the two end-member images
showing the least and most amount of total surface-water extent for each path/row, respectively.
Surface-water extent was then summed across the PPR and NP path/rows and divided by the
total area to calculate the hectares of surface-water extent per km$^2$ for each region. The NP
portion of path 27, row 30 (p27r30) and p30r30 were deleted, as was the PPR portion of p26r30
to avoid double counting overlapped path/rows.

**2.4 Stage One – Derivation of the Surface Water Climate Variable (SWCR)**

In stage one, surface-water extent in each HUC8 was related, using linear regression, to
water availability, defined as precipitation minus PET summed over a time interval. Water
availability provided an estimate of the amount of water in each watershed available to either (1)
runoff, (2) infiltrate to shallow or deep groundwater sources, or (3) be stored as surface-water.
Surface water was again partitioned into CCW and DCW using its spatial relationship to the
NHD. Precipitation data were compiled using the Parameter-elevation Regressions on
Independent Slopes Model (PRISM, Daly et al., 2008). PET, or the atmospheric demand for
evaporation and transpiration in the absence of water limitations, which can be expected over
open surface water, was compiled using gridded surface meteorological data PRISM and the
North American Land Data Assimilation System Phase 2 (Abatzoglou et al., 2011). PET was
calculated using the Penman-Monteith equation that required inputs of minimum and maximum
temperature, daily average dewpoint temperature (equivalently, vapor pressure or vapor pressure





deficit), wind speed and downward shortwave radiation (Abatzoglou et al., 2011, Mitchel et al.,
2004). The datasets were resampled to 125 m using cubic convolution and summarized for each
HUC8. Water availability was summed for a series of monthly periods preceding each image
date (3, 6, 9, 12, 18, 24, 30 and 36 months) to identify the accumulation period for which the
greatest number of HUC8's showed a significant ($p<0.05$) slope between water availability and
surface-water extent. This logic was meant to reduce the probability that a zero slope resulted
from surface water responding more strongly to climate drivers at a different time interval. This
first stage produced surface water climate response (SWCR), our dependent variables for stage 2,
i.e., the slope of the relationship between CCW and DCW surface-water extent to accumulated
water availability (Figure 2). The slope or stage 2 dependent variable is referred to as the surface
water climate response (SWCR) from this point forward.

Cloud cover makes it challenging to restrict analysis of Landsat imagery to a specific

season, while including imagery that covers more than one season potentially conflates seasonal
surface-water dynamics with interannual surface-water dynamics. The influence of seasonal
change in surface-water extent within our analysis contributed to the uncertainty (primarily
through sampling error) in the SWCR. For example, if we included an image from June 1993 and
one from August 1993 and related both images to the last nine months of precipitation and PET
(Sept 1992 - May 1993 and November 1992 – July 1993, respectively), greater seasonal
dynamics or variation in surface-water extent between the two dates can be expected to show up
as greater uncertainty in the slope, defined by the standard error of the slope or standard error of
the SWCR. This becomes more evident as the accumulated period becomes larger (e.g., 36
months). By explicitly considering the uncertainty of the SWCR in the regression analysis, as



described below in the Stage 2 Analysis (Section 2.6), we can, to the extent possible, account for
seasonally induced variation in surface-water extent.

**2.5 Landscape Variables for Stage 2 Analysis**

The independent variables summarized for each HUC8 and included in the analysis were

selected to characterize mechanisms through which water can leave the landscape (e.g.,
infiltration, runoff, tile drainage), mechanisms through which water can remain and expand on
the landscape (e.g., wetland density, wetland size, topography), as well as other potential
influences on surface water dynamics (e.g., climate norms, land cover). The National Wetland
Inventory (USFWS, 2010) and NHD stream dataset (USGS, 2013) were used to calculate
wetland and stream characteristics including stream density, wetland count and areal density, and
proportion of total wetland area attributed to large (>8 ha) features. A threshold of 8 hectares was
selected as this is the size threshold used by USFWS to define a lacustrine system (Cowardin et
al., 1979). We do not refer to these features as lakes, however, as water depth and associated
vegetation are also important features to defining lacustrine systems, and were not evaluated. We
did not include distance variables, which were previously found to be highly correlated with
simpler variables already in the analyses: mean wetland-to-wetland distance was previously
found to be highly correlated with wetland density (r = -0.95, p<0.01) and mean wetland-to-
stream distance highly correlated with stream density (r = 0.88, p<0.01) (Vanderhoof et al.,
2017). Surface topography can influence the capacity for surface water to expand and was
quantified as the weighted averaged slope gradient, as defined by the U.S. Department of
Agriculture's Soil Survey Geographic (SSURGO) Database (Soil Survey Staff, 2017).
Topographic Wetness Index was not included because of the relative weakness of such indices in



landscapes with little relief (e.g., Schmidt and Persson, 2003) and the data intensive nature of
calculating TWI with a 10 m DEM across such a large study area. Additional variables derived
from the SSURGO database to characterize infiltration capacity include available water storage
(0 - 150 cm), annual minimum depth to water table, and saturated hydraulic conductivity (Ksat).
Human influence was quantified as the abundance of agricultural activities, or the percent of
each HUC8 classified as agriculture, defined as the NLCD (2011) cover categories hay/pasture
and row crop. Anthropogenic modifications to drainage systems, or the percent land cover
artificially drained, was estimated as the percent of each HUC8 where row crop cover type
(NLCD 2011) and very poorly drained or poorly drained soils as defined by the National
Resources Conservation Service's SSURGO database were collocated following Christensen et
al., (2013). The climate normals per HUC8 (1989-2013) were calculated to represent the Landsat
image range. The precipitation averages are provided as part of the PRISM dataset (Daly et al.,
2008). PET was calculated as a function of monthly mean PRISM temperature and day length
following Hamon (1961). The Moisture Index (MI) was calculated as the ratio of precipitation
and PET where, if PET exceeded precipitation, MI = precipitation/PET – 1, and if precipitation
exceed or equaled PET, then MI = 1 = PET/precipitation. Values range from -1 (dry) to 1 (wet)
(Willmott and Feddema, 1992; Feddema, 2005). The climate averages were resampled to 1 km
from 4 km using inverse-distance weighting, prior to being averaged per HUC8. The distribution
of values within each of the independent variables are shown in Table 4. Spearman rank
correlations with a Bonferroni correction (Dunn, 1961) were calculated for the independent
variables (Table 5).

**2.6 Stage 2 - Analysis - Landscape Mechanisms Explaining Variability in SWCR**

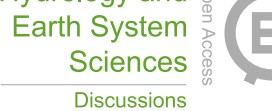



In stage two, CCW and DCW SWCRs, or the slope of the relationship between CCW and
DCW and accumulated water availability, were related to landscape variables using feasible
generalized least-squares (FGLS) regression, with HUC8s (n=150) as the unit of analysis. FGLS
allowed us to estimate the heteroscedastic structure of the residuals (Lewis and Linzer, 2005) and
has been previously applied within landscape ecology contexts (e.g., Acharya, 2000; Villalobos-
Jimenéz and Hassall, 2017). The SWCRs were found to be significant for the largest number of
HUC8s using a 9-month period of accumulation for both CCW and DCW, which was therefore
used as the accumulation period for further analyses (Table 6). The SWCRs were found to be
spatially autocorrelated using Global Moran's I (spatial relationship conceptualized using inverse
distance) (DCW SWCR, 9 months, z-score=7.8, p<0.01, CCW SWCR, 9 month, z-score=4.1,
p<0.01), violating the assumption of independence between samples. To account for spatial
autocorrelation in the SWCRs, we calculated an autocovariate in ArcGIS 10.3, Geostatistical
Analyst (ESRI, Redmond CA) which uses adjacent HUC8s to create a neighbor value. By
including a spatial autocovariate in the ordinary least-squares (OLS) regression model, we
controlled for how much the response variable reflected response values of adjacent HUCs,
before identifying additional significant explanatory variables (Dormann et al., 2007; Betts et al.,
2009). The autocovariate was automatically retained while only significant independent variables
(p<0.05) were additionally retained. The dependent variable was normalized using a Box-Cox
power transformation (R package MASS, Venables and Ripley, 2002). Multicollinearity was
formally assessed using the regression collinearity diagnostics described by Belsley et al. (1980)
and implemented in the R package perturb (Hendrickx, 2012). Collinearity may affect parameter
estimation when a condition index greater than 10 is associated with variance decomposition



proportions greater than 0.5 for two or more explanatory variables (Belsley, 1991). Both models
complied with collinearity requirements.

Having an estimated dependent variable (e.g., SWCR) does not necessarily present a

problem for a regression analysis, but we must recognize that the regression model error term
contains two components: (1) the expected random error resulting from sources of variation not
accounted for in the model, and (2) the difference between the true value of the dependent
variable and the estimated value (sampling error). In this study, the uncertainty around the
dependent variable (SWCR) was not constant across observations. Instead, the dependent
variable showed a strong positive correlation with its standard error (DCW SWCR, $R^2 = 0.59$,
$p<0.05$; CCW SWCR, $R^2 = 0.70$, $p<0.05$) (Figure 3). FGLS allowed us to estimate both
components of the error. To do so, we: (1) calculated the logarithm of squared residuals from the
OLS model, (2) regressed the log-residuals on the independent variables included in the OLS
model, (3) calculated the exponential of fitted values from that regression, which estimates the
variance of the regression residual that is not due to sampling of the dependent variable, $z$, and
(4) estimated the basic model again now including weights $(1\ z^{-1})$ (Hanushek, 1974; Lewis and
Linzer, 2005). We found the final model residuals to be random using the studentized Breusch-
Pagan test (Breusch and Pagan, 1979).

To help add confidence regarding which landscape variables were more or less important,

we also fit random forest models in R using the package randomForest (Liaw and Wiener, 2015).
The random forests were run with the SWCRs as the dependent variable and landscape
characteristics as independent variables. We derived 500 binary trees or bootstrap iterations
using out of bag (OOB) samples (70% of samples to train and 30% of samples to validate).
Variable importance was calculated as the change in node impurity (i.e., Gini importance).



Random forest models are generally insensitive to collinearity among metrics, however the
inclusion of correlated variables can deflate variable importance as well as the overall variation
explained by the model (Murphy et al., 2010). We implemented random forest model selection to
select the smallest number of non-redundant variables (varSelRF R package) (Murphy et al.,

2010).


**3 Results**
**3.1 Surface-Water Extent**
Median surface-water extent as well as the amount of water added and lost from the
surface between wet and dry periods was found to vary considerably across the study area
(Figure 4 and 5). Analysis of the median total surface-water extent between the PPR and the NP
demonstrated that the PPR had 2.6 times greater surface-water extent than the NP (Table 7). The
PPR also showed greater variability in total surface-water extent, adding 5.7 ha km$^{-2}$ during very
wet conditions and losing 2.8 ha km$^{-2}$ during very dry conditions, for a maximum net difference
of 8 ha km$^{-2}$. This can be compared to the NP which gained 1.6 ha km$^{-2}$ during very wet
conditions and lost 0.8 ha km$^{-2}$ during very dry conditions, a net difference of 2.4 ha km$^{-2}$ (Table
7). DCW, or water that was discontinuous with the stream network, showed greater expansion
and contraction in extent in both the PPR and NP, relative to CCW which intersected the stream
network. Consequently, DCW increased as a percent of total surface water during wet periods
and decreased as a percent of total surface water in dry periods. This suggests that across the
study area, surface water that was disconnected from the stream network disproportionately
served a surface water storage function during wet periods, reducing the amount of water



contributing to downstream flooding. Similarly, DCWs disproportionately experienced loss
during dry periods.

**3.2 Relationship between Surface-Water Extent and Water Availability**

Including PET instead of using precipitation alone tended to increase the percentage of

HUC8s showing a statistically significant relationship between surface-water extent and water
availability across the different accumulation periods that we tested, although this was not true
for all time periods. For instance, the percent change from precipitation to precipitation minus
PET ranged from -1.4 to 38% for DCW and -6.3 to 24.3% for CCW. For DCW there was a jump
in the percentage of HUC8s showing a significant relationship between six and nine months, but
the percentage of HUC8s stabilized after this time period out to 36 months. CCW showed a
similar but smaller jump in the percentage of HUC8s with a significant relationship between six
and nine months (Table 6). At nine months, all images, regardless of being collected in the
spring, summer or fall, would include winter precipitation. We observed substantial spatial
variability in the statistical relationship between surface-water extent and water availability.
Using nine months as the accumulation period, we observed a strong spatial pattern in DCW.
PPR HUC8s tended to show a greater SWCR, exhibited by a substantial increase in surface-
water extent with increased water availability, while HUC8s across the NP tended to show a
smaller SWCR, exhibited by minor to no increases in surface-water extent with increased water
availability (Figure 6 and 7). For CCW, the spatial pattern was less consistent within the PPR or
ecoregion boundaries. Instead, HUC8s with a greater SWCR tended to be HUC8s with large
lakes or floodplains (Figure 6 and 7).





### 3.3 Landscape Variables Explaining Variability in Surface-Water Response


For DCW SWCR, when independent variables were assessed individually using
Spearman's rank correlation, the SWCR was greater in locations with fewer streams ($R = -0.64$,
$p<0.05$), lower slope gradient ($R = -0.59$, $p<0.05$), higher wetland density ($R = 0.52$, $p<0.05$) and
total wetland area ($R = 0.51$, $p<0.05$), deeper minimum depth to water table ($R = 0.59$, $p<0.05$)
and where a greater proportion of the total surface water was disconnected from the stream
network ($R = 0.42$, $p<0.05$) (Table 8). When the relative importance of the variables was tested
using random forest, variables found to be the most important included, wetland density, stream
density, annual minimum depth to water table and the slope gradient (Table 8). However, after
accounting for the spatial autocorrelation in the DCW SWCR and the significance of the
variables, the DCW SWCR increased in the final feasible generalized least-squares model
(adjusted $R^2 = 0.66$, F-statistic = 73.6) with (1) greater wetland density, (2) deeper depth to
groundwater, and (3) less anthropogenic drainage (Table 9). The variable most consistent
identified across statistical approaches was wetland density.
For CCW SWCR, fewer independent variables showed a significant Spearman rank
correlation. The SWCR for stream-connected water increased in locations with a greater total
wetland area ($R = 0.48$, $p<0.05$) and less average precipitation ($R = -0.33$, $p<0.05$) (Table 8).
Using random forest, the total wetland area and proportion of total water from large features
were found to be the most important variables in explaining variation. The final feasible
generalized least-squares model (adjusted $R^2 = 0.54$, F-statistic = 37.4) also found the
relationship between CCW and surface-water availability (i.e., SWCR) was stronger with greater
total wetland area, but also found that it decreased with greater wetland density (Table 9).



## 4. Discussion


Surface-water extent, and in particular surface water within well-studied portions of the

PPR, has been previously shown to exhibit seasonal and interannual patterns (Poff et al., 1997;
Beeri and Phillips, 2007; Vanderhoof et al., 2016) that can, in turn, influence the cumulative
hydrologic response of a watershed (Golden et al. 2016; Evenson et al. 2016; Ali and Creed
2017). What has been less studied is how surface-water dynamics vary across diverse
landscapes. This is particularly relevant when we consider the need for communities and local
agencies to plan ahead for expected changes in the precipitation regime associated with climate
change (Dore, 2005; Johnson et al., 2005; Millett et al., 2009). Our results demonstrated that the
relationship between surface-water extent and water availability (SWCR) is a function of both
climate and landscape variables and that the density of depressional wetlands, in particular,
played a key explanatory role in the observed landscape response to increased climate inputs.
Given our findings, we expect that changes in net precipitation from climate change or other
climatic forcings will disproportionately affect surface-water extent across the PPR relative to
the adjacent NP, and that these effects will be more evident in disconnected wetland systems
(DCWs) than in wetlands connected to the river network (CCWs). Surface waters that are
disconnected from the stream network showed a larger change in extent in response to wetter
conditions in landscapes with higher wetland densities. That is to say that landscapes with a
larger number of depressional features were found to show a greater increase in surface-water
extent in response to a wetter climate, relative to landscapes with fewer depressional features. In
landscapes with more concentrated water, greater total wetland area, but lower wetland density,
surface waters connected to the stream network showed more substantial expansion with
increased water availability. This finding suggests that the presence of stream-connected lakes





within large flat basins may be an important factor influencing surface-water expansion.
Previous research found lakes within the PPR to be important features that commonly experience
extensive surface-water expansion, subsuming adjacent wetlands during wet periods
(Vanderhoof and Alexander, 2016). These findings suggest that if climate conditions within the
U.S. portion of the PPR continue to get wetter, as predicted (e.g., Millett et al. 2009), then both
small wetland depressions and larger features, such as lakes and floodplains, will both serve
critical roles in storing increased inputs of surface water, which could prevent downstream
flooding.
Our study area was intentionally selected to encompass a large area with a wide range of
landscape conditions in regards to wetland and stream density and capacity for infiltration.
Across the study area, variation in the values of many of the variables (e.g., stream density,
wetland density) can be attributed to landscape age or the time since the last glacial retreat, and
corresponding variability in drainage development across the region (Ahnert, 1996). The
Wisconsin glacier retreated from the PPR by 11,300 BP, meaning the drainage system is still
developing and surface water is being stored in glacially formed depressions (Winter and
Rosenberry, 1998; Stokes et al., 2007). In contrast, the landscape to the west and south of the
PPR, is much older (>20,000 BP) with a well-developed drainage network (Clayton and Moran,

1982).

In addition to extensive human-induced wetland loss across the region (Miller et al.,
2009; Van Meter et al., 2015), the drainage network across the region is also increasingly
modified with the expansion of ditch networks and tile drainage in association with agricultural
activities (McCauley et al., 2015). Ditches, pipes and field tiles on the glacial till can hasten the
speed with which water leaves a location and lower the water table through increased water



withdrawal (De Laney, 1995; Blann et al., 2009; McCauley et al., 2015). We found in the FGLS
model, the expansion of disconnected water was inversely related to the abundance of estimated
anthropogenic drainage. Because anthropogenic drainage increases the rate at which water leaves
a location, it results in the loss or reduction of landscape-scale functions of wetlands and other
natural water storage features in the PPR (McCauley et al. 2015), and shifts the hydrologic
behaviors of watersheds towards those more commonly seen in the NP.

Evapotranspiration is known to be a primary mechanism for water loss in the PPR

(Winter and Rosenberry, 1998). By explicitly incorporating this value into the SWCR, we could
better isolate the effects of landscape-based components such as surface storage, stream density,
and topography. One challenging component to characterize was the capacity for water to
infiltrate through soil horizons. Depth to bedrock SSURGO data was found to be too patchy (i.e.,
too much missing data) to be useful. A variable that instead was found to correlate significantly
with the expansion of disconnected water was annual minimum depth to groundwater. The PPR
tended to have a deeper minimum depth while the NP had a shallower minimum depth, on
average. A reduction in infiltration due to the low permeability of glacial till (Sloan, 1972;
Winter and Rosenberry, 1995), would reduce the potential for increased water table elevations.
Concomitantly, with less infiltration, pulses of snowmelt or precipitation in the PPR would
instead be transported as overland flow and fill wetlands with available storage.

We must also consider that we may be missing key landscape variables that could explain

variability in the spatial response of surface-water extent to climate inputs. For example, major
landscape characteristics required for stream-connected surface water to expand include (1)
large, stream-connected water bodies such as lakes and (2) hydrologically-connected floodplains.
The influence of large water bodies was considered by including total wetland area and the





portion of water from larger (>8 ha) features, however we did not explicitly consider the
presence/absence of active floodplains beyond including stream density as a variable. Floodplain
activity typically exhibits strong seasonal patterns; however, the goal of our analysis was to
focus on patterns of surface-water extent that occurred on longer-time scales (i.e., interannual
variability). Because of this, we excluded two Landsat path/rows from the analysis that were
originally included because strong seasonal flooding outweighed interannual patterns in climate
as evidenced by a lack of a relationship between climate indices (e.g., Standardized Precipitation
Index (12 months) and Palmer Hydrologic Drought Index) and surface-water extent. These
path/rows included p30r27 which straddles North Dakota and Minnesota and exhibits strong
seasonal flooding of the Red River and p28r32 in the southeastern corner of Nebraska, which
exhibits strong seasonal flooding of the Missouri River. However, even with the exclusion of
these two path/rows, the importance of floodplains is still evident in Figure 6B where we
observed greater SWCR in areas with an abundance of lakes or floodplain systems. Because
complete floodplain maps across the study area are lacking, we were not able to explicitly
identify the role of floodplains in the CCW models.

In addition to decision points regarding study area extent, other decision points may have

influenced our findings. For example, the period of time for which the greatest number of
HUC8s showed a significant SWCR was used as the climate accumulation period. This logic was
meant to avoid, to the extent possible, a HUC8 showing a zero SWCR because surface water
responded at a time period different than the one selected. However, its usage meant that the
study results are limited to interpreting the relationship of surface-water extent to same year
climate inputs (or the previous 9 months) and may be less applicable to understanding the
relationship of surface-water extent to shorter (seasonal) or longer (multi-year) time periods. In





addition, decisions regarding image inclusion may have also influenced the analysis. Although
the Landsat images used in the analysis were selected strategically to represent historically dry,
average, and wet conditions, because the Landsat images were processed individually we were
ultimately limited in the number of Landsat images we could process. As more remotely sensed
products become available, such as the U.S. Geological Survey's Dynamic Surface Water Extent
(DSWE) Product, which plans to utilize the entire Landsat archive (1984 to present) (Jones,
2015), we could utilize many more images and reduce the uncertainty in estimates of the SWCR
or watershed-specific response to available water. Although decision points regarding the data
included or excluded from the analysis are important to consider, this study provides an
improved understanding of how the relationship between surface-water extent and climate may
vary spatially across different landscapes.

**5. Conclusion**

Shifts in climate patterns and the frequency of extreme climate events will influence

surface-water extent. This has implications for habitat availability (Boschilia et al., 2008;
Calhoun et al., 2017), agricultural productivity (Mokrech et al., 2008; Gornall et al., 2010) and
hydrologic connectivity (Golden et al. 2016; Ali and Creed 2017). This study demonstrated that
not only is surface-water extent variable across landscapes, but shifts in climate patterns will
have an uneven effect on surface-water extent across these different landscapes. The PPR
experienced a 2.6 fold greater surface-water extent than the adjacent NP under average
conditions and a 3.4 fold larger range in surface-water extent between drought and deluge
conditions. To move from ecoregion boundaries to a more functional characterization of the
spatial distribution of surface water on the landscape, we used a statistical approach to explore



potentially significant landscape variables that could explain the spatially variable change in
surface water to climate inputs (precipitation minus evapotranspiration). Landscapes with higher
wetland density and less anthropogenic drainage showed a greater expansion of disconnected
(from the stream network) surface water (e.g., depressional wetlands) and wetter climatic
conditions relative to landscapes with fewer wetlands and more anthropogenic drainage.
Landscapes with fewer wetlands but more total surface water area (e.g., lakes, large river
systems) showed a greater expansion of stream-connected surface water and wetter climatic
conditions relative to landscapes with less total surface water area. Enhancing our knowledge of
spatial and temporal variability in the relationship between surface-water extent and climate
inputs can advance efforts to predict the hydrologic effects of climate change, including drought
and floods, on water resources and improve hydrological modeling in low gradient landscapes.

**Acknowledgements**
This research was funded by the Drought Resilience Initiative through an interagency agreement
with the U.S. Environmental Protection Agency, Office of Research and Development (DW-014-
92454401 - 0). We thank Tedros Berhane, Hayley Distler, Marena Gilbert and Clifton Burt for
their assistance in processing the Landsat imagery and ancillary datasets. We thank Maliha Nash
for providing data on climate averages. The views expressed in this manuscript are solely those
of the authors and do not necessarily reflect the views or policies of the U.S. EPA. Any use of
trade, firm, or product names is for descriptive purposes only and does not imply endorsement by
the U.S. Government.

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





**Tables**
**Table 1.** A summary of the Landsat images utilized within each selected path/row. Landsat TM images were used for dates 2011 and
earlier. Landsat 8 OLI images were used for 2013 forward. DOY: day of year; PHDI: Palmer Hydrological Drought Index. *p37r26
was considered NP because of its dissimilarity with the rest of the PPR.

| Path/Row | PPR/Northern Prairie (NP) (primary) | Number of Images | Spring (DOY 60-151) | Summer (DOY 152-243) | Fall (DOY 244-335) | Year Range | Min. PHDI (%) | Max. PHDI (%) | Mean PHDI (%) |
|---|---|---|---|---|---|---|---|---|---|
| p26r30 | NP | 12 | 6 | 4 | 2 | 1987-2010 | 4 | 99 | 45 |
| p26r32 | NP | 17 | 10 | 3 | 4 | 1988-2010 | 2 | 99 | 51 |
| p27r30 | PPR | 9 | 3 | 4 | 2 | 1988-2008 | 4 | 99 | 54 |
| p29r29 | PPR | 17 | 9 | 2 | 6 | 1990-2011 | 7 | 100 | 69 |
| p30r30 | PPR | 13 | 5 | 5 | 3 | 1988-2013 | 2 | 100 | 45 |
| p30r31 | NP | 15 | 6 | 5 | 4 | 1986-2011 | 5 | 94 | 38 |
| p31r27 | PPR | 15 | 2 | 6 | 7 | 1990-2011 | 3 | 100 | 67 |
| p31r29 | PPR | 13 | 6 | 5 | 2 | 1989-2011 | 7 | 99 | 45 |
| p33r28 | NP | 15 | 8 | 2 | 5 | 1988-2015 | 1 | 99 | 49 |
| p36r28 | NP | 16 | 7 | 7 | 2 | 1985-2013 | 2 | 96 | 38 |
| p37r26 | NP* | 15 | 4 | 6 | 5 | 1987-2013 | 1 | 99 | 52 |
| **Total** | | **157** | **66** | **49** | **42** | | | | |






**Table 2.** Landsat images and corresponding National Agricultural Imaging Program (NAIP) images used to validate the Landsat surface-water extent maps. Accuracy is presented here by Landsat image. PHDI: Palmer Hydrological Drought Index, SP12: 12 month Standardized Precipitation Index, OE: omission error for water, CE: commission error for water, OA: overall accuracy, DC: Dice coefficient, RB: relative bias

| Landsat Path/Row | Landsat date | NAIP date(s) | Gap (days) | PHDI | SP12 | Number of points | OE (%) | CE (%) | OA (%) | DC (%) | RB (%) |
|---|---|---|---|---|---|---|---|---|---|---|---|
| p26r32 | 28-Jun-04 | 23-Jun-04 and 07-Jul-04 | -5 to +9 days | 0.57 | 0.14 | 947 | 6.3 | 5.9 | 97.4 | 93.9 | -0.5 |
| p27r30 | 14-Jul-13 | 10-Jul-13 and 12-Jul-13 | -4 to -2 days | -0.34 | 0.05 | 707 | 11.8 | 9.3 | 92.5 | 89.5 | -2.7 |
| p29r29 | 13-Oct-06 | 25-Sep-06 | -18 days | 2.3 | -0.08 | 814 | 11.1 | 2.5 | 93.6 | 93.0 | -8.8 |
| p29r29 | 8-Oct-10 | 17-Sep-10 and 20-Sep-10 | +18 to +21 days | 9.63 | 3.06 | 959 | 1.9 | 3.3 | 97.4 | 96.4 | 1.4 |
| p31r29 | 17-Jul-04 | 10-Jul-04 and 14-Jul-04 | -7 to -3 days | -0.4 | -0.04 | 1302 | 7.4 | 1.5 | 97.2 | 95.4 | -6.0 |
| p33r28 | 13-Jul-03 | 11-Jul-03 and 15-Jul-03 | -2 to +2 days | -2.74 | -0.91 | 908 | 10.6 | 27.0 | 85.5 | 80.4 | 22.5 |
| p37r26 | 31-Jul-11 | 16-Jul-11 and 19-Jul-11 | -15 to -12 days | 2.96 | 1.29 | 498 | 16.8 | 9.7 | 90.2 | 86.6 | -7.9 |

**Table 3.** Summary of accuracy statistics across all of the Landsat images validated using National Agricultural Imaging Program (NAIP) imagery.

| | NAIP - Inundated | NAIP - Non-Inundated | Total |
|---|---|---|---|
| Landsat - Inundated | 2052 | 183 | 2235 |
| Landsat - Non-Inundated | 190 | 3710 | 3900 |
| Total | 2242 | 3893 | 6135 |
| | | | |
| Omission error for water (%) | 8.5 | | |
| Commission error for water (%) | 8.2 | | |
| Overall Accuracy (%) | 93.9 | | |
| Dice Coefficient | 91.7 | | |
| Relative Bias | 0.0 | | |



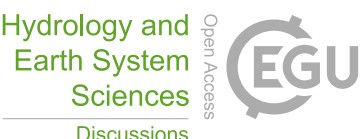

**Table 4.** Independent variables considered in the landscape analysis and the distribution of values for each variable across the 8-digit hydrological units (HUC8s). Mean values for the HUC8s within the Prairie Pothole Region (PPR) and Northern Prairie (NP) are also shown with significant differences (p<0.01) between the two groups, as determined by the Wilcoxon rank sum test, indicated by different superscript letters. NHD: National Hydrography Dataset, NWI: National Wetland Inventory, PRISM: Parameter-elevation Regressions on Independent Slopes Model, SSURGO: Soil Survey Geographic Database, NLCD: National Land Cover Database, DCW: disconnected surface water, PET: potential evapotranspiration, avg: average

| Independent Variables | Units | Range | 25th % | 50th % | 75th % | PPR (avg) | NP (avg) | Source |
|---|---|---|---|---|---|---|---|---|
| **Wetland and Stream Characteristics** | | | | | | | | |
| Stream density | m ha$^{-1}$ | 0.1 to 26.1 | 7.2 | 11.4 | 15.0 | 7.8[a] | 14.5[b] | High-Resolution NHD (USGS 2013) |
| Total wetland density | no ha$^{-1}$ | 0 to 0.2 | 0.02 | 0.03 | 0.06 | 0.06[a] | 0.03[b] | NWI (USFWS 2010) |
| Total wetland areal abundance | ha ha$^{-1}$ | 0 to 0.7 | 0.02 | 0.03 | 0.08 | 0.08[a] | 0.05[b] | NWI (USFWS 2010) |
| Portion of total water area from large features | % | 0.1 to 97.8 | 32.1 | 44.7 | 58.0 | 45.0[a] | 47.2[a] | NWI (USFWS 2010, Zhang et al., 2009) |
| Portion DCW of total surface water | % | 0 to 100 | 10.6 | 24.4 | 50.0 | 44.5[a] | 22.8[b] | Landsat and NHD (USGS 2013) |
| **Climate Averages** | | | | | | | | |
| Moisture Index Average | ~ | -0.4 to 0.7 | -0.1 | -0.04 | 0.2 | 0.04[a] | -0.03[a] | PRISM (Daly et al., 2008) |
| Precipitation Average | mm yr-1 | 312.3 to 1007.8 | 490.3 | 599.6 | 790.8 | 641.5[a] | 624.3[a] | PRISM (Daly et al., 2008) |
| PET Average | mm yr-1 | 496.2 to 683.0 | 564.2 | 595.5 | 628.9 | 595.5[a] | 594.8[a] | PRISM (Daly et al., 2008) |
| **Soil and Topography** | | | | | | | | |
| Available water storage (0-150 cm), weighted | cm | 7.6 to 29.5 | 18.0 | 22.8 | 24.7 | 24.0[a] | 19.1[b] | SSURGO (Soil Survey Staff, 2017) |
| Annual minimum depth to water table | cm | 0.1 to 69.0 | 11 | 24.8 | 43.3 | 40.5[a] | 17.9[b] | SSURGO (Soil Survey Staff, 2017) |
| Ksat | μm sec$^{-1}$ | 2.1 to 107.7 | 8.4 | 13.8 | 22.5 | 21.4[a] | 21.2[a] | SSURGO (Soil Survey Staff, 2017) |
| Slope gradient, weighted average | % | 1.5 to 19.2 | 3.0 | 4.3 | 7.1 | 3.3[a] | 7.1[b] | SSURGO (Soil Survey Staff, 2017) |
| **Human Influence** | | | | | | | | |
| Agricultural land cover | % | 0.1 to 92.0 | 25.2 | 62.8 | 80.5 | 72.6[b] | 39.6[b] | 2011 NLCD (Homer et al., 2015) |
| Percent drained by anthropogenic means | % | 0 to 93.0 | 5.9 | 50.1 | 77.8 | 61.7[a] | 32.5[b] | 2011 NLCD and SSURGO |





**Table 5.** Spearman rank correlation values between the independent variables considered in the analysis. Bonferonni correction was applied to the p-values and significant correlations (p<0.05) are starred. DCW: surface water disconnected from the stream network, CCW: continuously connected surface water, MI: Moisture Index, PET: potential evapotranspiration, precip: precipitation, lg: large, ag: agricultural, Ksat: saturated hydraulic conductivity, na: not applicable

| Variable | DCW auto-covariate | CCW auto-covariate | Portion dis-connected | Stream density | Wetland density | Wetland areal abund. | Dominance of lg. water bodies | MI | Precip | PET | Avail water storage (0-150 cm) | Annual min depth to water table | Ksat | Slope gradient | Ag land cover | Percent drained |
|---|---|---|---|---|---|---|---|---|---|---|---|---|---|---|---|---|
| DCW autocovariate | 1 | | | | | | | | | | | | | | 0.33* | 0.22 |
| CCW autocovariate | na | 1 | | | | | | | | | | | | | -0.03 | -0.07 |
| Portion DCW of total water | 0.45* | -0.11 | 1 | | | | | | | | | | | | 0.46* | 0.26 |
| Stream density | -0.66* | -0.16 | -0.38* | 1 | | | | | | | | | | | -0.33* | -0.2 |
| Wetland density | 0.48* | 0.15 | 0.32* | -0.33* | 1 | | | | | | | | | | 0.11 | 0.25 |
| Wetland areal abundance | 0.48* | 0.27 | -0.09 | -0.37* | 0.79* | 1 | | | | | | | | | 0.05 | 0.23 |
| Dominance of lg water bodies | -0.04 | 0.18 | -0.63* | -0.05 | -0.02 | 0.44 | 1 | | | | | | | | -0.24 | -0.01 |
| MI | 0.03 | -0.29 | 0.33* | -0.34* | 0.24 | 0.18 | -0.22 | 1 | | | | | | | 0.80* | 0.64* |
| Precipitation | -0.05 | -0.26 | 0.20 | -0.21 | 0.19 | 0.06 | -0.16 | 0.86* | 1 | | | | | | 0.66* | 0.50* |
| PET | 0.03 | 0.15 | -0.05 | 0.09 | -0.1 | -0.1 | 0 | -0.2 | 0.25 | 1 | | | | | -0.08 | -0.16 |
| Avail water storage (0-150 cm) | 0.29 | 0.05 | 0.37* | -0.34* | 0.26 | 0.21 | -0.11 | 0.60* | 0.48* | -0.07 | 1 | | | | 0.66* | 0.51* |
| Annual min depth to water table | 0.54* | -0.04 | 0.54* | -0.62* | 0.29 | 0.26 | -0.1 | 0.67* | 0.44* | -0.21 | 0.49* | 1 | | | 0.69* | 0.57* |
| Ksat | 0.21 | 0.01 | 0.11 | -0.47* | -0.03 | -0.01 | 0.08 | 0.14 | 0.03 | -0.2 | -0.05 | 0.19 | 1 | | 0.02 | -0.07 |
| Slope gradient | -0.58* | -0.16 | -0.34* | 0.66* | -0.19 | -0.29 | 0.1 | -0.41* | -0.17 | 0.12 | -0.44* | -0.61* | -0.25 | 1 | -0.63* | -0.32* |
| Agricultural land cover | 0.33* | -0.03 | 0.46* | -0.33* | 0.11 | 0.05 | -0.24 | 0.80* | 0.66* | -0.08 | 0.66* | 0.69* | 0.02 | -0.63* | 1 | 0.63* |





**Table 6.** The percent of HUC8s across the study area that showed a significant relationship (p<0.05) between surface-water extent and (1) precipitation (Precip or P) or (2) precipitation minus potential evapotranspiration (PET) for different accumulation periods. DCW: disconnected surface water; CCW: continuously, connected surface water.

| Accumulated Period | Precip DCW (%) | P - PET DCW (%) | Inclusion of PET change (DCW) | Precip CCW (%) | P - PET CCW (%) | Inclusion of PET change (CCW) |
|---|---|---|---|---|---|---|
| 3 months | 19.4 | 27.1 | 7.6 | 15.3 | 28.5 | 13.2 |
| 6 months | 5.6 | 31.9 | **26.4** | 9.0 | 33.3 | **24.3** |
| 9 months | 20.8 | **59.7** | **38.9** | 27.1 | **48.6** | **21.5** |
| 12 months | 45.8 | 50.7 | 4.9 | 42.4 | 41.0 | -1.4 |
| 18 months | 24.3 | **58.3** | **34.0** | 25.7 | 39.6 | 13.9 |
| 24 months | **52.1** | 50.7 | -1.4 | **43.8** | 37.5 | -6.3 |
| 30 months | 28.5 | 55.6 | **27.1** | 27.1 | 43.1 | 16.0 |
| 36 months | **54.9** | 54.9 | 0.0 | **47.2** | **44.4** | -2.8 |
| HUC8s with a sig relationship in at least 1 time period | 65.3 | 75.7 | 10.4 | 59.0 | 67.4 | 8.3 |







**Table 7.** Surface-water extent conditions summarized for the Prairie Pothole Region (PPR) and adjacent Northern Prairie (NP). TSW: total surface-water extent, CCW: continuously connected surface water that intersects the stream network, DCW: disconnected surface water or surface water that does not directly intersect the stream network.

| Region | Path/rows (all or part) | Total area (km²) | Min (ha km⁻²) | Max (ha km⁻²) | Median (ha km⁻²) | Added min to max (ha km⁻²) | Reduction from median to min (%) | Increase from median to max (%) | Min (% of all) (area) | Max (% of all) (area) | Median (% of all) (area) |
|---|---|---|---|---|---|---|---|---|---|---|---|
| PPR TSW | 7 | 146,309 | 3.51 | 11.99 | 6.33 | 8.48 | 44.6 | 89.2 | ~ | ~ | ~ |
| NP TSW | 9 | 173,026 | 1.62 | 4.07 | 2.45 | 2.45 | 33.9 | 66.1 | ~ | ~ | ~ |
| PPR CCW | 7 | 146,309 | 2.82 | 7.56 | 4.44 | 4.74 | 36.5 | 70.4 | 80.3 | 63.1 | 70.1 |
| NP CCW | 9 | 173,026 | 1.44 | 3.11 | 2.06 | 1.66 | 30.0 | 50.5 | 89.1 | 76.3 | 84.2 |
| PPR DCW | 7 | 146,309 | 0.69 | 4.42 | 1.90 | 3.73 | 63.4 | 133.4 | 19.7 | 36.9 | 29.9 |
| NP DCW | 9 | 173,026 | 0.18 | 0.97 | 0.39 | 0.79 | 54.4 | 149.2 | 10.9 | 23.7 | 15.8 |





**Table 8.** Spearman rank correlation values between the dependent variables and each of the independent variables considered in the analysis. Bonferonni correction was applied to the p-values and significant correlations (p<0.05) are starred. Relative variable importance as determined by random forest models are also presented for each variable (i.e., increase in node purity). PET: potential evapotranspiration, Ksat: saturated hydraulic conductivity, DCW: disconnected surface water, CCW: continuously, connected surface water

| Variable | Response (DCW, 9 months) | | Response (CCW, 9 months) | |
|---|---|---|---|---|
| | Spearman rank correlation | Increase in node purity | Spearman rank correlation | Increase in node purity |
| Autocovariate | 0.79* | 0.081 | 0.53* | 0.108 |
| Portion DCW is of total surface water | 0.42* | 0.012 | -0.11 | 0.334[I] |
| Stream density | -0.64* | 0.036[I] | -0.15 | 0.060 |
| Wetland density | 0.52* | 0.048[I] | 0.27 | 0.057[I] |
| Wetland areal abundance | 0.51* | 0.017[I] | 0.48* | 0.855[I] |
| Portion of total water from large features | -0.01 | 0.004 | 0.30 | 0.556[I] |
| Moisture Index (average) | -0.03 | 0.005 | -0.28 | 0.053[I] |
| Precipitation (average) | -0.10 | 0.008[I] | -0.33* | 0.039[I] |
| PET (average) | -0.06 | 0.011[I] | -0.13 | 0.034 |
| Available water storage (0-150 cm) | 0.27 | 0.007 | -0.01 | 0.061 |
| Annual minimum depth to water table | 0.56* | 0.027[I] | 0.09 | 0.046 |
| Ksat | 0.04 | 0.004 | -0.08 | 0.070[I] |
| Slope gradient, weighted average | -0.59* | 0.025[I] | -0.22 | 0.072 |
| Agricultural land cover | 0.31 | 0.005 | -0.05 | 0.035 |
| Percent drained by anthropogenic means | 0.22 | 0.004 | -0.04 | 0.020 |

[I]Variables selected by the random forest model selection process, using the R package rfUtilities, when the autocovariate was not included.







**Table 9.** Feasible generalized least square models with residual weights applied relating the response (of surface-water extent to water availability) to landscape-related variables. All variables included in the models were significant. DCW: surface water disconnected from the stream network, CCW: continuously connected surface water, SE: standard error, D.F.: degrees of freedom

| Response of DCW water to water availability | Variables | Coefficients | SE | t-value |
|---|---|---|---|---|
| D.F. = 145 | Intercept | 0.17 | 0.01 | 12.84 |
| F-statistic = 73.6 | Autocovariate | 0.03 | 0.004 | 6.32 |
| adjusted $R^2$ = 0.66 | Wetland density | 0.90 | 0.23 | 3.96 |
| | Minimum depth to groundwater | 0.0021 | 0.0006 | 3.29 |
| | Percent anthropogenically drained | -0.0004 | 0.0003 | -1.25 |

| Response of CCW water to water availability | Variables | Coefficients | SE | t-value |
|---|---|---|---|---|
| D.F. = 144 | Intercept | 0.018 | 0.01 | 1.43 |
| F-statistic = 69.4 | Wetland areal abundance | 0.96 | 0.07 | 14.42 |
| adjusted $R^2$ = 0.58 | Wetland density | -0.43 | 0.21 | -2.09 |
| | Autocovariate | -0.12 | 0.01 | -0.89 |






**Figures**

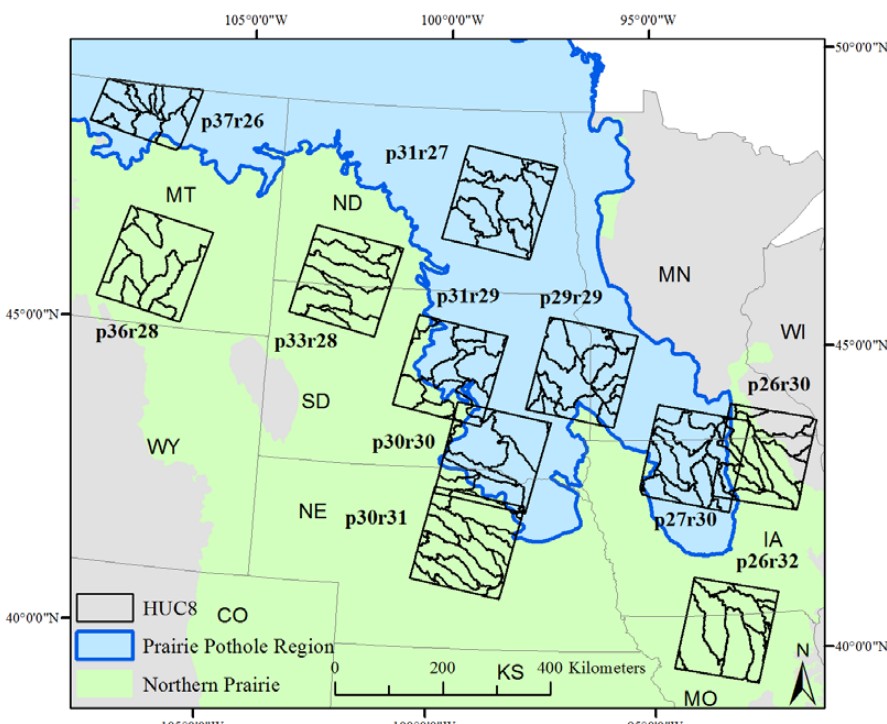


**Figure 1.** Distribution of Landsat path/rows used to map surface-water extent and corresponding
8-digit Hydrological Units (HUC8s) used for further analysis in relation to the boundary of the
Prairie Pothole Region (PPR). The p37r26 behaved dissimilarly from the PPR and similarly to
the adjacent Northern Prairie (NP) in all regards and was therefore included with the NP for
analyses organized by PPR and NP.





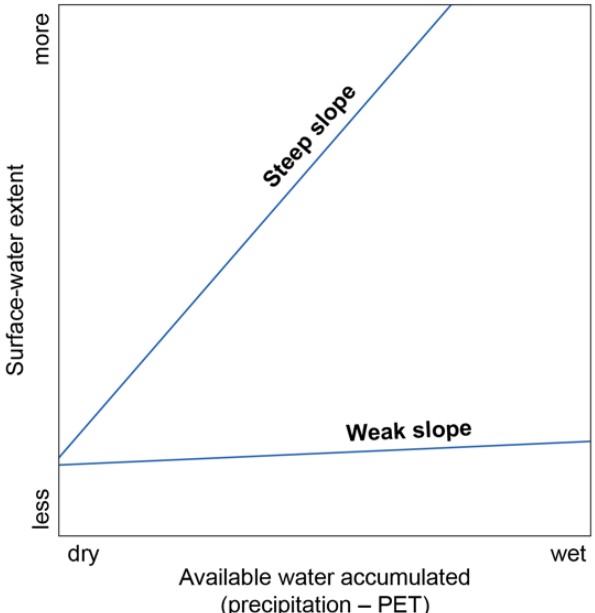

**Figure 2.** Theoretical figure showing the derived dependent variable, or the Surface Water Climate Response (SWCR), defined as the slope of the statistical relationship between accumulated water and surface-water extent. Some areas show a greater SWCR or substantial increase in surface-water extent as more water becomes available via precipitation minus potential evapotranspiration (PET), while other areas show little to no change in surface-water extent, presumably as excess water leaves the system through runoff or infiltration.

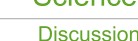
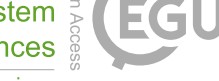

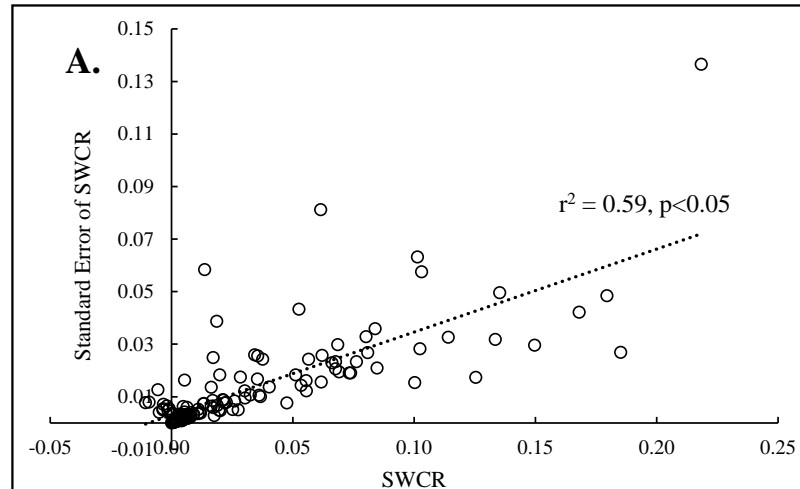

955

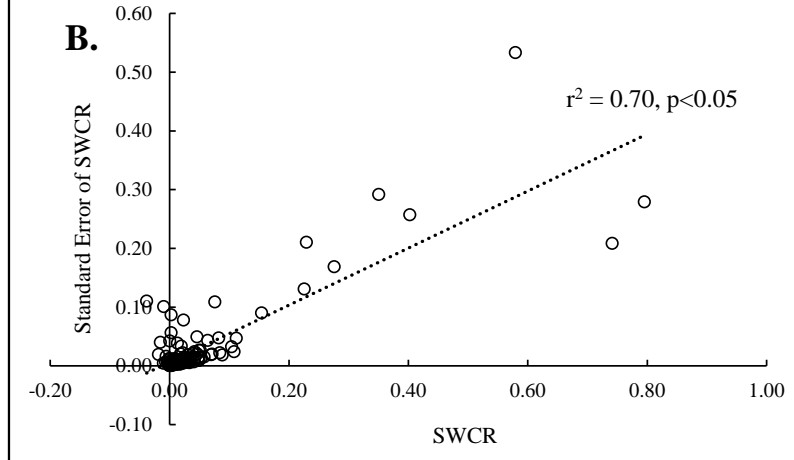

956

**Figure 3.** Standard errors of the Surface Water Climate Response (SWCR) tended to be
positively correlated with both A) discontinuous surface water (DCW) or surface water
disconnected from the stream network and B) continuously connected water (CCW) or surface
water connected to the stream network.

961





962

**Figure 4.** Mean surface-water abundance and the amount of "wetting up" varied substantially between different Landsat path/rows. Portions of the Northern Prairie (e.g., p26r30) showed relatively less surface-water extent and expansion (A and B) while portions of the Prairie Pothole Region (e.g., p29r29) showed relatively more surface-water extent and expansion (C and D). Note: not all water is visible at this zoomed-out scale. PHDI: Palmer Hydrological Drought Index

969







**Figure 5.** Examples of minor and substantial expansion of surface-water extent between historically dry and historically wet points in time. PHDI: Palmer Hydrological Drought Index.

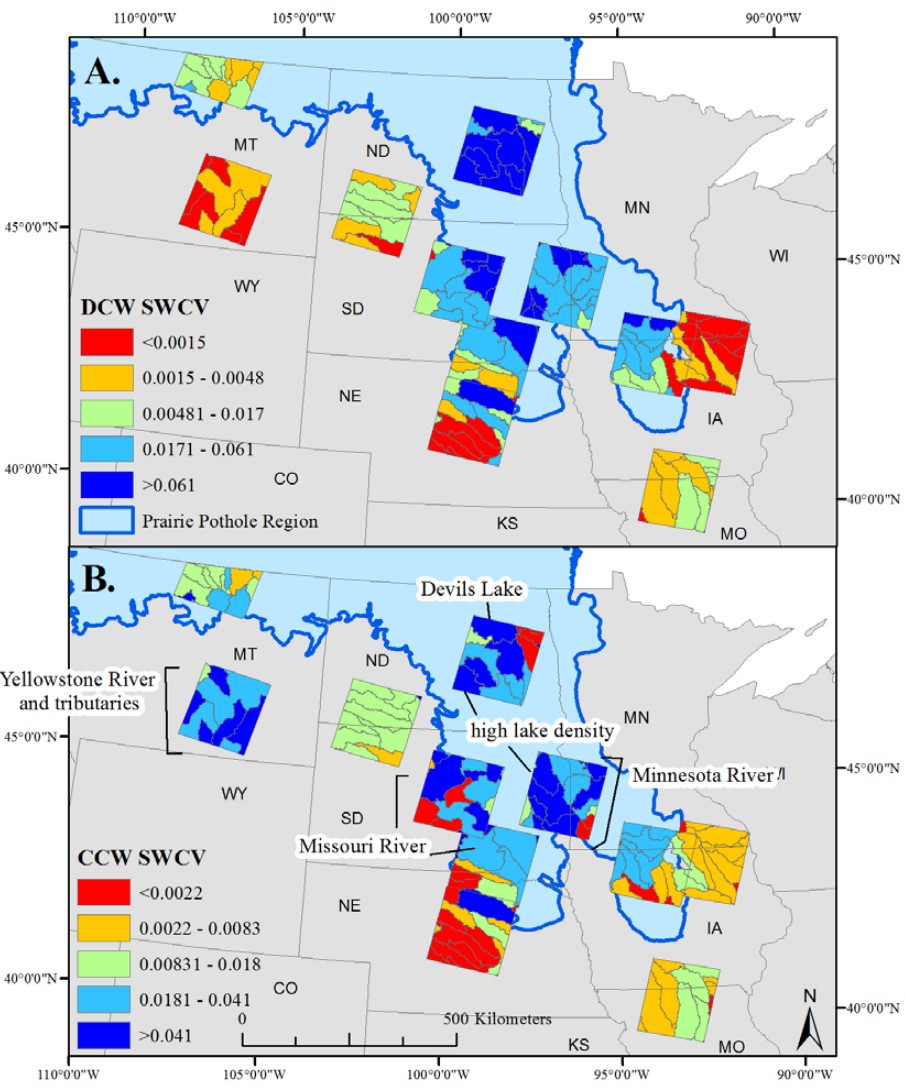

973

**Figure 6.** The spatial distribution of the Surface Water Climate Variable (SWCR) values from
the statistical relationships between available water, defined as precipitation minus potential
evapotranspiration accumulated over the previous 9 months, and surface-water extent. Greater
SWCR values indicate greater change in surface-water extent with increased available water.
Surface-water extent was divided between A) disconnected surface water (DCW), or surface-
water extent disconnected from the stream network, and B) continuously connected water
(CCW), or surface-water extent connected to the stream network.

981



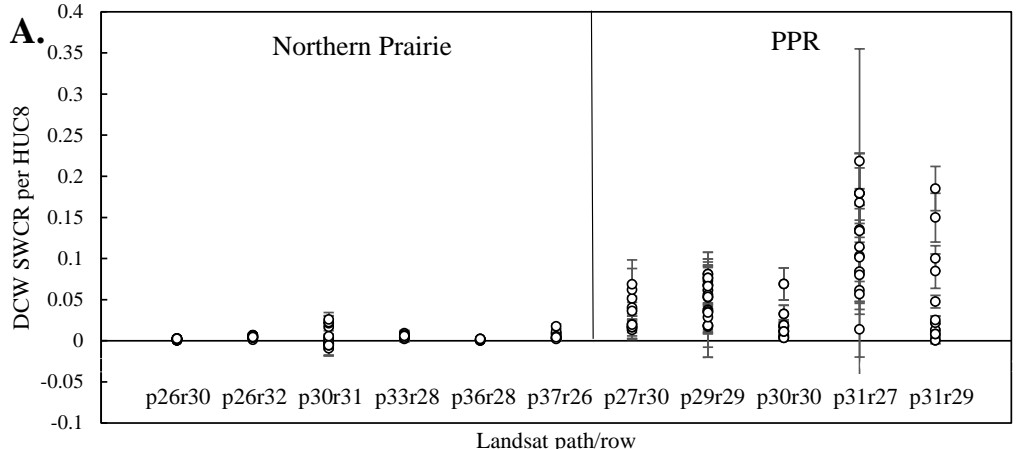

982

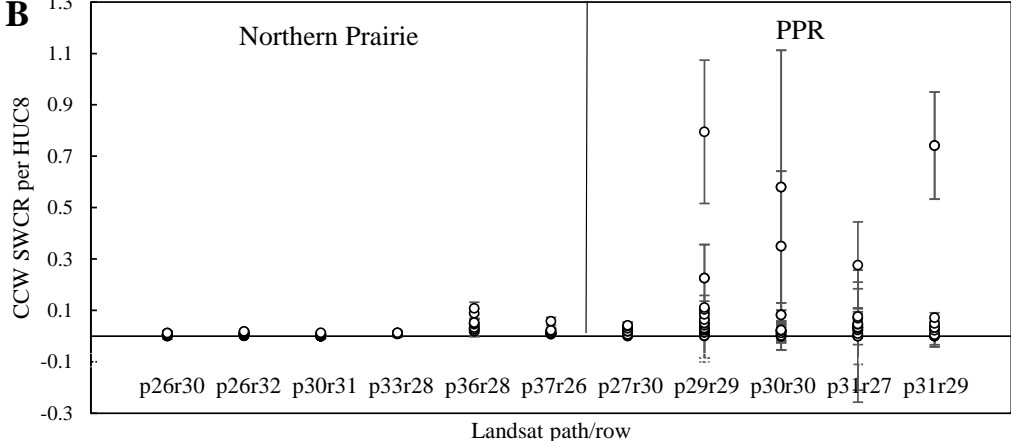

983

**Figure 7.** Distribution of Surface Water Climate Response and standard error values organized
by Landsat path/row and primary path/row location, i.e., the Northern Prairie or the Prairie
Pothole Region (PPR) for A) surface water that is disconnected from the stream network (DCW),
and B) surface water that is connected to the stream network (CCW). HUC8: 8-digit
Hydrological Units







**Appendix**
**Table 1.** A complete list of Landsat TM images used in the analysis and the corresponding
Palmer Hydrological Drought Index (PHDI).

| Landsat path/row | Date | PHDI | Landsat path/row | Date | PHDI | Landsat path/row | Date | PHDI |
|---|---|---|---|---|---|---|---|---|
| p26r30 | 1989 170 | -4.29 | p30r30 | 1990 121 | -4.70 | p31r29 | 1989 109 | -1.62 |
| p26r30 | 1989 186 | -4.29 | p30r30 | 1989 294 | -4.66 | p31r29 | 2003 196 | -1.22 |
| p26r30 | 1988 296 | -4.15 | p30r30 | 1989 110 | -3.47 | p31r29 | 2004 279 | 2.52 |
| p26r30 | 1996 222 | -0.24 | p30r30 | 1991 236 | -2.79 | p31r29 | 1999 121 | 5.19 |
| p26r30 | 1987 117 | 0.06 | p30r30 | 1988 148 | -1.23 | p31r29 | 2011 154 | 6.55 |
| p26r30 | 1996 142 | 0.30 | p30r30 | 2002 122 | -1.12 | p31r29 | 2010 167 | 6.94 |
| p26r30 | 2010 148 | 1.10 | p30r30 | 2013 184 | -0.94 | p31r29 | 2010 279 | 8.63 |
| p26r30 | 2006 153 | 1.17 | p30r30 | 2003 141 | 0.26 | p33r28 | 1988 249 | -5.68 |
| p26r30 | 2008 95 | 2.82 | p30r30 | 2003 285 | 0.88 | p33r28 | 1990 254 | -3.87 |
| p26r30 | 1993 133 | 3.95 | p30r30 | 1993 161 | 5.40 | p33r28 | 2008 112 | -2.86 |
| p26r30 | 1993 277 | 6.92 | p30r30 | 2011 211 | 6.49 | p33r28 | 1988 137 | -2.47 |
| p26r32 | 1988 264 | -4.18 | p30r30 | 2011 179 | 6.87 | p33r28 | 2005 135 | -2.35 |
| p26r32 | 2000 105 | -3.03 | p30r30 | 2010 288 | 8.93 | p33r28 | 2003 146 | -1.78 |
| p26r32 | 2003 145 | -2.98 | p30r31 | 2002 250 | -4.62 | p33r28 | 2005 263 | -0.62 |
| p26r32 | 1989 266 | -2.92 | p30r31 | 2000 269 | -3.75 | p33r28 | 1998 148 | 0.22 |
| p26r32 | 1991 288 | -1.88 | p30r31 | 2000 173 | -2.66 | p33r28 | 2006 106 | 0.36 |
| p26r32 | 1991 96 | 0.55 | p30r31 | 1990 105 | -2.63 | p33r28 | 1998 260 | 0.70 |
| p26r32 | 2007 108 | 0.74 | p30r31 | 2003 141 | -2.46 | p33r28 | 1995 188 | 4.09 |
| p26r32 | 2002 158 | 1.59 | p30r31 | 1990 297 | -2.45 | p33r28 | 1997 129 | 5.11 |
| p26r32 | 1994 136 | 2.76 | p30r31 | 1990 137 | -2.43 | p33r28 | 2015 67 | 5.37 |
| p26r32 | 1993 133 | 3.66 | p30r31 | 2003 221 | -2.41 | p33r28 | 2014 160 | 5.61 |
| p26r32 | 1994 104 | 3.79 | p30r31 | 2000 221 | -2.38 | p33r28 | 2014 256 | 9.15 |
| p26r32 | 2010 100 | 4.06 | p30r31 | 2000 125 | -2.05 | p36r28 | 1988 222 | -6.07 |
| p26r32 | 2008 271 | 5.07 | p30r31 | 2002 122 | -1.84 | p36r28 | 2002 212 | -5.14 |
| p26r32 | 2010 228 | 5.90 | p30r31 | 2005 178 | 1.58 | p36r28 | 2004 154 | -4.72 |
| p27r30 | 1988 239 | -4.52 | p30r31 | 1986 174 | 2.19 | p36r28 | 2004 282 | -4.29 |
| p27r30 | 1989 161 | -4.34 | p30r31 | 1994 148 | 3.63 | p36r28 | 2003 135 | -2.38 |
| p27r30 | 2003 280 | -1.32 | p30r31 | 1994 260 | 4.12 | p36r28 | 1985 149 | -2.04 |
| p27r30 | 2002 141 | -1.25 | p30r31 | 2011 179 | 5.22 | p36r28 | 1989 112 | -1.94 |
| p27r30 | 2003 104 | 1.44 | p30r31 | 2009 173 | 5.29 | p36r28 | 2013 178 | -0.91 |
| p27r30 | 2008 182 | 3.03 | p31r27 | 1990 160 | -4.12 | p36r28 | 1993 91 | -0.89 |
| p27r30 | 1992 266 | 3.22 | p31r27 | 2006 252 | -3.32 | p36r28 | 2013 242 | -0.42 |
| p27r30 | 1992 122 | 4.29 | p31r27 | 1991 163 | -2.45 | p36r28 | 1998 121 | 1.67 |
| p27r30 | 1993 172 | 6.52 | p31r27 | 1992 118 | -1.93 | p36r28 | 2008 181 | 1.70 |
| p29r29 | 1990 130 | -3.55 | p31r27 | 1999 121 | 2.01 | p36r28 | 1996 244 | 2.06 |
| p29r29 | 2003 118 | -2.01 | p31r27 | 2007 255 | 2.41 | p36r28 | 1996 100 | 3.81 |
| p29r29 | 2002 323 | -1.69 | p31r27 | 1997 195 | 2.72 | p36r28 | 1993 235 | 5.17 |
| p29r29 | 1991 133 | -0.69 | p31r27 | 2005 169 | 3.06 | p36r28 | 1994 142 | |
| p29r29 | 1992 136 | 1.35 | p31r27 | 2009 244 | 3.28 | p37r26 | 1988 213 | -5.70 |
| p29r29 | 2006 286 | 2.30 | p31r27 | 2004 279 | 4.38 | p37r26 | 2006 246 | -3.41 |
| p29r29 | 1998 120 | 2.77 | p31r27 | 2001 190 | 4.46 | p37r26 | 1994 261 | -2.54 |
| p29r29 | 2005 91 | 3.15 | p31r27 | 1995 270 | 5.97 | p37r26 | 2008 108 | -2.37 |
| p29r29 | 2006 94 | 4.20 | p31r27 | 2010 279 | 6.43 | p37r26 | 2002 171 | -1.85 |
| p29r29 | 2001 128 | 4.47 | p31r27 | 2011 186 | 6.61 | p37r26 | 1991 141 | 0.14 |
| p29r29 | 1997 165 | 5.05 | p31r27 | 1994 299 | 7.03 | p37r26 | 2009 142 | 0.26 |
| p29r29 | 1995 288 | 5.71 | p31r27 | 2011 266 | 8.92 | p37r26 | 1995 168 | 1.35 |





| p29r29 | 2011 284 | 5.88 | p31r29 | 2006 172 | -3.49 | p37r26 | 1995 264 | 1.68 |
|--------|----------|------|--------|----------|-------|--------|----------|------|
| p29r29 | 2010 105 | 6.19 | p31r29 | 1989 189 | -3.38 | p37r26 | 1987 162 | 2.15 |
| p29r29 | 1993 266 | 6.86 | p31r29 | 2004 135 | -2.66 | p37r26 | 1991 269 | 2.26 |
| p29r29 | 2011 156 | 8.37 | p31r29 | 1989 269 | -2.31 | p37r26 | 1994 101 | 2.76 |
| p29r29 | 2010 281 | 9.63 | p31r29 | 2003 100 | -2.24 | p37r26 | 2013 169 | 3.40 |
|        |          |      | p31r29 | 2003 132 | -1.84 | p37r26 | 2011 276 | 7.32 |
|        |          |      | p31r29 | 1990 96  | -1.65 | p37r26 | 2011 212 | 9.14 |


