# Peer review of "Wetlands inform how climate extremes influence surface water expansion and contraction"

_Hydrology and Earth System Sciences, 2017_

## Referee Comment (RC1) · Anonymous Referee #1 · 8 Dec 2017

General Comments

Overall, the authors address in interesting comparison in how differences in geomorphology can influence landscape surface-water responses in different ecoregions. This paper is well written and important for the field of wetland ecohydrology in the Midwestern USA. The analytical methods and statistical tools show a compelling story that the PPR contains a higher concentration of depressional basins than the NP and therefore surface water in the PPR responds very strongly to changes in climate. Most of my suggestions are areas where the authors can clarify and citations they can add to give the reader a better understanding of climate shifts in the region.

[Figure]

Specific Comments

Your paper alludes to other studies that looked at the relationship between surface water and climate, but you do not cite a recent paper from the PPR. It would be helpful to cite this paper especially in your discussion about shifts in climate patterns: McKenna, O.P., Mushet, D.M., Rosenberry, D.O., LaBaugh, J.W. Evidence for a climate-induced ecohydrological state shift in wetland ecosystems of the southern Prairie Pothole Region. Climatic Change (2017) 145: 273. https://doi.org/10.1007/s10584-017-2097-7

L363-373 please clarify why climate variables are included in stage 2 of the analysis, I would think that they would be in the first stage for developing the SWCR

L471-472 how is a metric regarding amount of surface area disconnected from stream network an independent variable? Isn't this overlapping with the definition of DCW?

I would like to see something in the discussion about table 7 regarding differences in DCW vs CCW area in NP and PPR. When controlling for wetland density are there significant differences between proportion of DCW vs CCW in NP as compared to PPR? This would help specify some of the discussion points in L501-511

Table 5 seems pretty raw and could be moved to appendix. Especially since Table 8 and Table 9 are giving more advanced analyses on significant independent variables

Is it fair to use the Missouri River in Fig 5 to represent PPR? At the very least you need to specify which examples came from PPR and which came from NP in fig. 5 legend. Missouri River seems to be the border between the two regions.

The final models from Table 9 need to be used more in the discussion especially building on how CCW and DCW responses may change in the face of climate and land-use change Why in Figure 6 are Yellowstone River and tributaries so responsive to climate as compared to other CCW and DCW sites in NP? Also, isn't Devils Lake naturally a DCW and it is only CCW because of pumping into Sheyenne River?

Technical corrections
L70-73 long and confusing sentence, consider re-wording or breaking up.

L541-552 This paragraph seems unnecessary. Either give more context or remove.

Fig 6 legend should read "DCW SWCR" and "CCW SWCR"

———————————————————

---

## Referee Comment (RC2) · Anonymous Referee #2 · 26 Dec 2017

General Comments:

The authors attempted to analyze the spatiotemporal variations of surface-water expansion and contraction across the Prairie Pothole Region (PPR) and the adjacent Northern Prairie (NP) of the United States using time-series Landsat images (1985-2015). By delineating the time-series surface-water extent, the authors investigated how landscape characteristics (infiltration capacity, surface storage capacity, stream density, etc.) influenced the relationships between climate inputs and surface-water dynamics differently in the PPR and NP. Overall, the manuscript is well written and it is a welcome contribution to the field of wetland hydrology in the Prairie Pothole Region

of North America. I have a few minor comments that might help improve the quality of the manuscript.

Specific Comments:

One of the major undertakings of this paper is mapping surface-water extent by classifying 157 Landsat images, which is a huge amount of effort. The authors stated that the image classification algorithm is trained on a water spectral signature, which was derived from open-water polygons manually selected within each path/row, resulting in a water signature specific to each image (see Lines 217-219). To make the research reproducible, I suggest the authors elaborate the manual delineation of open-water polygons for deriving water spectral signature. For example, what's the minimum size of polygons? On average, how many polygons were manually delineated for each Landsat image? Did the Landsat images with the same path/row use the same open-water polygons?

It seems the authors did not mention the minimum wetland/surface-water size they were trying to map. To my knowledge, the median size of PPR wetlands is less than 2000 m2, which is approximately equal to the size of two Landsat pixels. On the one hand, image objects with only a few pixels might not be reliable classification results. On the other hand, small wetlands (< 2 pixels) might be more sensitive to climate change. How would the minimum size of wetlands influence the regression results?

Lines 291-293: How about p31r29? This Landsat scene also lies across both PPR and NP.

Table 2 shows that the overall accuracy for p33r28 is 85.5%, which is significantly lower than other Landsat images (90∼97%). I think this deserves some explanation.

Appendix Table 1: It would make more sense to me if the Landsat images of each path/row are listed in a chronological order of image acquisition dates. I would also suggest adding a dashed line to separate different path/row (e.g., between p26r30 and

p26r32), which can make this long table a bit easier to read. I also noticed that the PHDI for p36r28-1994-142 is missing. Why?

It would increase the impact of this paper and benefit the community if the authors can make the surface-water mapping products available to the public.

Technical Corrections:

Lines 226/338: National Wetland Inventory -> National Wetlands Inventory

Line 227: "Select images"?

Lines 892/897: National Agricultural Imaging Program -> National Agricultural Imagery Program

---

## Author Comment (AC1) · 13 Feb 2018

Reviewer #1: General Comments: Overall, the authors address an interesting comparison in how differences in geomorphology can influence landscape surface-water responses in different ecoregions. This paper is well written and important for the field of wetland ecohydrology in the Midwestern USA. The analytical methods and statistical tools show a compelling story that the PPR contains a higher concentration of depressional basins than the NP and therefore surface water in the PPR responds very strongly to changes in climate. Most of my suggestions are areas where the authors can clarify and citations they can add to give the reader a better understanding of

climate shifts in the region. Response: Thank you for your thoughtful comments which are addressed below.

Specific Comments Comment: Your paper alludes to other studies that looked at the relationship between surface water and climate, but you do not cite a recent paper from the PPR. It would be helpful to cite this paper especially in your discussion about shifts in climate patterns: McKenna, O.P., Mushet, D.M., Rosenberry, D.O., LaBaugh, J.W. Evidence for a climate-induced ecohydrological state shift in wetland ecosystems of the southern Prairie Pothole Region. Climatic Change (2017) 145: 273. https://doi.org/10.1007/s10584-017-2097-7 L363-373 Response: We have added this reference as recommended.

Comment: please clarify why climate variables are included in stage 2 of the analysis, I would think that they would be in the first stage for developing the SWCR. Response: To clarify, in Stage 1 we related Precipitation – Potential Evapotranspiration (PET) (aggregated over the previous 9 months) to inundation, so climate variables were directly used to derive the SWCR. Only multidecadal climate normals (averaged over 1989-2013) were used as independent variables in the stage 2. We added the following text to clarify, "Multi-decadal climate normals were included to test for the potential effect of a climate gradient across the study area."

Comment: L471-472 how is a metric regarding amount of surface area disconnected from stream network an independent variable? Isn't this overlapping with the definition of DCW? Response: We apologize for the confusion. It is the proportion (%) of DCW water, so the variable is attempting to get at whether watershed storage is dominated by disconnected wetlands or connected wetlands. As discontinuous waters are often small, depressional wetlands, they may or may not comprise a substantial amount of the total storage capacity across the watershed. We added the following text to clarify, "We included the proportion (%) DCW was of total surface water as a proxy of the relative distribution of water storage across the watershed between riparian and non-riparian water bodies."

Comment: I would like to see something in the discussion about table 7 regarding differences in DCW vs CCW area in NP and PPR. When controlling for wetland density are there significant differences between proportion of DCW vs CCW in NP as compared to PPR? This would help specify some of the discussion points in L501-511. Response: We do continue to see more water being added to DCW even after controlling for wetland density. We have added further discussion of this to the Discussion section.

Comment: Table 5 seems pretty raw and could be moved to appendix. Especially since Table 8 and Table 9 are giving more advanced analyses on significant independent variables Response: We have moved Table 5 to the Appendix (now Table A2) and correspondingly renumbered the remaining tables.

Comment: Is it fair to use the Missouri River in Fig 5 to represent PPR? At the very least you need to specify which examples came from PPR and which came from NP in fig. 5 legend. Missouri River seems to be the border between the two regions. Response: Figure 5 was meant to show the difference in patterns of expansion between DCW (wetland density) and CCW (lakes and floodplains). It was not mean to represent the PPR vs NP. To clarify this we have added several new references to Figure 5 in the text to indicate this.

Comment: The final models from Table 9 need to be used more in the discussion especially building on how CCW and DCW responses may change in the face of climate and land-use change Response: We have modified the Discussion section, especially its organization, and in particular the Conclusion section to more adequately address this comment, in particular how responses relate to climate and land-use change.

Comment: Why in Figure 6 are Yellowstone River and tributaries so responsive to climate as compared to other CCW and DCW sites in NP? Also, isn't Devils Lake naturally a DCW and it is only CCW because of pumping into Sheyenne River? Response: These are good questions. In regards to Yellowstone River and its tributaries, I suspect
the climate signal was clear because this path/row had relatively low wetland density (see Figure 6A), and the rivers were of such a size that as they started to fill up/widen, they began to be more consistently mapped by Landsat. However, this is mostly speculation so I haven't added this to the text. In regards to Devils Lake, you are correct, however we used the intersection of water with the NHD lines to define stream connected consistently. We recognize that in certain cases, this means stream lines may or may not connect to downstream waters.

Comment: L70-73 long and confusing sentence, consider re-wording or breaking up. Response: We broke this sentence into 2 sentences.

Comment: L541-552 This paragraph seems unnecessary. Either give more context or remove. Response: We heavily modified this paragraph and better contextualized it with the model results. As annual minimum depth to water table was a significant variable in the DCW SWCR model we feel that it is important to retain discussion of this variable.

Comment: Fig 6 legend should read "DCW SWCR" and "CCW SWCR" Response: We have updated the figure as recommended.

---

## Author Comment (AC2) · 13 Feb 2018

Reviewer # 2

General Comments: The authors attempted to analyze the spatiotemporal variations of surface-water expansion and contraction across the Prairie Pothole Region (PPR) and the adjacent Northern Prairie (NP) of the United States using time-series Landsat images (1985- 2015). By delineating the time-series surface-water extent, the authors investigated how landscape characteristics (infiltration capacity, surface storage capacity, stream density, etc.) influenced the relationships between climate inputs and surface-water dynamics differently in the PPR and NP. Overall, the manuscript is well

written and it is a welcome contribution to the field of wetland hydrology in the Prairie Pothole Region I have a few minor comments that might help improve the quality of the manuscript. Response: Thank you for your thoughtful comments which are addressed below.

Specific Comments: Comment: One of the major undertakings of this paper is mapping surface-water extent by classifying 157 Landsat images, which is a huge amount of effort. The authors stated that the image classification algorithm is trained on a water spectral signature, which was derived from open-water polygons manually selected within each path/row, resulting in a water signature specific to each image (see Lines 217-219). To make the research reproducible, I suggest the authors elaborate the manual delineation of open-water polygons for deriving water spectral signature. For example, what's the minimum size of polygons? On average, how many polygons were manually delineated for each Landsat image? Did the Landsat images with the same path/row use the same openwater polygons? Response: Additional text has been added to expand on the selection of training polygons. "Three to four polygons (minimum size of 1 ha per polygon, total training area per path/row was approximately 20 ha) per path/row were selected. The same open-water polygons were used to train the time series for each path/row."

Comment: It seems the authors did not mention the minimum wetland/surface-water size they were trying to map. To my knowledge, the median size of PPR wetlands is less than 2000 m2, which is approximately equal to the size of two Landsat pixels. On the one hand, image objects with only a few pixels might not be reliable classification results. On the other hand, small wetlands (< 2 pixels) might be more sensitive to climate change. How would the minimum size of wetlands influence the regression results? Response: We agree that the small median size of PPR wetlands truly presents a challenge for remotely sensed analysis at a landscape scale. We have added a new analysis to the validation section in which we randomly selected 400 NWI wetlands (from <0.1 ha to 1.0 ha) visibly inundated in the NAIP imagery. Wetlands larger than

0.2 ha were reliably detected (73%), which is better than most efforts using Landsat imagery (minimum wetlands size is typically 0.8 to 1.0 ha). We have also added text to the Discussion section explaining this source of uncertainty.

Lines 291-293: How about p31r29? This Landsat scene also lies across both PPR and NP. Response: The NP and PPR portions of p31r29 were analyzed separately. We have added this text to the Methods section.

Table 2 shows that the overall accuracy for p33r28 is 85.5%, which is significantly lower than other Landsat images (90âĹij97%). I think this deserves some explanation. Response: The higher commission error in p33r28 can be attributed to confusion with bare rock which is abundant in the northwest portion of the path/row as well as uncertainty across agricultural fields. We added the following text, "Errors of commission were higher for p33r28 which can be attributed to confusion in agricultural fields and with bare rock formations."

Appendix Table 1: It would make more sense to me if the Landsat images of each path/row are listed in a chronological order of image acquisition dates. I would also suggest adding a dashed line to separate different path/row (e.g., between p26r30 and p26r32), which can make this long table a bit easier to read. I also noticed that the PHDI for p36r28-1994-142 is missing. Why? Response: We have made all changes to the Appendix Table 1 as recommended.

Comment: It would increase the impact of this paper and benefit the community if the authors can make the surface-water mapping products available to the public. Response: We agree, supporting USGS Data Policies, the Landsat surface-water maps will be published in ScienceBase (https://www.sciencebase.gov/catalog/), following the article's publication.

Technical Corrections: Lines 226/338: National Wetland Inventory -> National Wetlands Inventory Line 227: "Select images"? Response: Changed as recommended.
Lines 892/897: National Agricultural Imaging Program -> National Agricultural Imagery Program Response: Changed as recommended.

---

## Author Comment (AC4) · 13 Feb 2018

Document showing response to comments and tracked changes is attached.

Please also note the supplement to this comment:
https://www.hydrol-earth-syst-sci-discuss.net/hess-2017-581/hess-2017-581-AC4-supplement.pdf
* * *